# Mechanical Characterization of the Elastoplastic Response of a C11000-H2 Copper Sheet

**DOI:** 10.3390/ma13225193

**Published:** 2020-11-17

**Authors:** Matías Pacheco, Claudio García-Herrera, Diego Celentano, Jean-Philippe Ponthot

**Affiliations:** 1Departamento de Ingeniería Mecánica, Universidad de Santiago de Chile, USACH, Av. Bernardo O’Higgins 3363, Santiago de Chile 9170124, Chile; matias.pacheco@usach.cl (M.P.); claudio.garcia@usach.cl (C.G.-H.); 2Departamento de Ingeniería Mecánica y Metalúrgica, Pontificia Universidad Católica de Chile, PUC, Av. Vicuña Mackenna 4860, Santiago de Chile 7820436, Chile; 3Department of Aerospace and Mechanical Engineering, University of Liège, Allée de la Découverte 13A, B-4000 Liège, Belgium; JP.Ponthot@uliege.be

**Keywords:** mechanical characterization, copper sheet, non-associated Hill-48 criterion, tensile test, bulge test

## Abstract

This work presents an elastoplastic characterization of a rolled C11000-H2 99.90% pure copper sheet considering the orthotropic non-associated Hill-48 criterion together with a modified Voce hardening law. One of the main features of this material is the necking formation at small strains levels causing the early development of non-homogeneous stress and strain patterns in the tested samples. Due to this fact, a robust inverse calibration approach, based on an experimental–analytical–numerical iterative predictor–corrector methodology, is proposed to obtain the constitutive material parameters. This fitting procedure, which uses tensile test measurements where the strains are obtained via digital image correlation (DIC), consists of three steps aimed at, respectively, determining (a) the parameters of the hardening model, (b) a first prediction of the Hill-48 parameters based on the Lankford coefficients and, (c) corrected parameters of the yield and flow potential functions that minimize the experimental–numerical error of the material response. Finally, this study shows that the mechanical characterization carried out in this context is capable of adequately predicting the behavior of the material in the bulge test.

## 1. Introduction

In the sheet metal forming industry, the creation of new parts is common for various devices such as, e.g., containers, pots, radiator components, cooling fins, and conductors. In general, these parts have intricate geometries and/or low dimensional tolerances and, consequently, the design of the forming process is fundamental for manufacturing a good quality component. Sheet forming processes are complex problems due mainly to the nonlinearity produced by friction, the development of large displacements and strains and, in addition, to the difficulty of modeling the anisotropic elastoplastic material behavior. Owing to this, the analysis of these processes is nowadays routinely tackled via numerical simulation using the finite element method [1,2,3,4]. In this context, an accurate elastoplastic modeling of the forming material is required in order to achieve an adequate description of its response during the deformation stages [2].

For this last purpose, a large number of models have been proposed to predict the evolution of hardening and plastic deformation during sheet forming operations. Hardening has been usually addressed with either the well-known Swift or Voce laws for materials exhibiting increasing or saturation hardening, respectively [5]. More recently, mixed Swift–Voce laws have been also proposed and used [6,7,8,9,10]. Regarding the material anisotropy generated by previous rolling operations, several orthotropic yield criteria have been proposed, e.g., the first and most widely used nowadays Hill-48 model and many other complex functions that are no longer quadratic [11,12,13,14,15,16,17]. Moreover, although the associated plastic flow rule is the most common choice in these works, there also are studies in which non-associated flow rules are used [18,19].

The parameter identification procedure of such models is usually performed via experimental data based on uniaxial tensile tests conducted on samples cut at different directions with respect to the rolling direction of the sheet. Yield limits together with parameters of the adopted hardening, yield, and flow potential functions are thus derived by minimizing the error between the experimental measurements and the model predictions in the stress–strain curves. A more robust characterization of the material response can be nowadays achieved by using an improved data acquisition system like the Digital Image Correlation (DIC) technique that allows obtaining the strain field of the sample during the whole deformation process [20,21]. Both the degree of complexity of the models, usually linked to their related number of material parameters, and the development of triaxial strain and stress patterns after the necking of the sample make the identification procedure via purely experimental–analytical approaches quite difficult. This drawback can be overcome via minimizing schemes involving numerical simulations of the complete deformation range up to the rupture stage of the sample. The use of this experimental–numerical characterization methodology is crucial in other experimental procedures that have been also considered to this end, e.g., hydraulic bulge [22], Taylor impact [23], spherical identation [24], and hydroforming [25] tests. The identification of parameters using a great variety of plastic anisotropic models have been extensively addressed for different steels [4,22,23,24,25] and many other materials such as aluminum [6,7,8,9,10,12,13,14,15,18,19,20,21], titanium [1], and magnesium alloys [3] and, in addition, high conductivity copper Cu-1/4H [11].

Despite the importance of copper in the manufacture of components, such as fins, batteries, cosmetics containers, and plumbing accessories, the characterization procedures described above have not focused on the modeling of the anisotropic elastoplastic response of nearly pure (i.e., >99.5%) copper sheets when they are subjected to forming operations. Although there are already some related works that reported on the validation of different yield criteria [11,26,27], the influence of anisotropy on different forming processes [28,29,30,31], and the grain effects in microforming [32,33,34], none of them were devoted to a comprehensive characterization and modeling of the elastoplastic behavior of this material at the macroscopic scale.

The aim of this work is the mechanical characterization of the large strain elastoplastic response of a rolled C11000-H2 99.90% pure copper sheet by means of the orthotropic non-associated Hill-48 criterion considering a modified Voce hardening law. To this end, the identification of the material parameters of this model is tackled by using the tensile stress–strain curves obtained from samples cut for three different directions with respect to the rolling direction of the sheet, here denoted as the rolling (RD), diagonal (DD), and transverse (TD) directions. In this context, the fitting procedure is usually carried out by minimizing the error between the model predictions and the corresponding experimental data. Due to the fact that necking occurs at very low strain levels (i.e., lower than 1% for samples stretched along DD and TD) for this material, a fitting procedure exclusively based on analytical expressions, which are only valid within the homogeneous strain range, is precluded in this case. Therefore, numerical simulations are needed to capture the complex triaxial stress and strain patterns that develop in the post-necking regime. Thus, the proposed inverse calibration approach, that takes into account the whole deformation range up to the rupture stage, consists in an experimental–analytical–numerical iterative predictor–corrector methodology including three steps intended to, respectively, determine (a) the parameters of the hardening model derived from the equivalent stress–strain curve of the RD sample; (b) a first prediction, assuming an associated plastic model, of the Hill-48 parameters based on the analytical equations of the RD, DD, and TD Lankford coefficients; and (c) corrected parameters of the yield and flow potential functions that minimize the experimental–numerical error of the material response given in terms of the engineering stress–strain and width–axial true strain curves for the RD, DD, and TD samples. In order to get accurate measurements of the strain field evolution during the tests, the DIC technique is used. Finally, the model parameters obtained with the presented mechanical characterization are assessed in the analysis of the material response in the bulge test by comparing numerical predictions with experimental data of the pressure-dome height curve and final thickness profiles along RD and TD.

## 2. Material and Methods

### 2.1. Experimental Procedure

#### 2.1.1. Material

The material used in this work is a 0.5 mm thick rolled C11000-H2 99.90% pure copper sheet. As this material is cold rolled, it is assumed to have an orthotropic behavior because the material grains have a preferred orientation after the rolling [35]. In this case, the commonly defined directions are the rolling (RD), diagonal (DD), transverse (TD), and normal (ND), as shown in Figure 1a.

#### 2.1.2. Tensile Test

The tensile test was performed following the ASTM standards [36]. The dimensions of the samples are depicted in Figure 1b, where a gradual reduction of the width (from 12.6 mm to 12.5 mm) that fits the standards has been considered to induce the formation of the necking in the center of the extensometer length. Three samples for each direction, i.e., RD, DD, and TD, were tested. The speed at which the tests were performed was 2 mm/min. As usual, the evolution of the axial forces and extensometer displacements were recorded during the test. In addition, the speckled pattern printed in the samples allowed to track the evolution of the displacement field through images obtained by DIC. The final setup of the test is shown in Figure 2.

The following parameters are obtained from the tensile test measurements: Young modulus *E*, yield strength $\sigma_{y}$, ultimate tensile strength $\sigma_{uts}$, Lankford coefficients *R*, strain at the onset of necking $\varepsilon_{neck}$, and fracture strain $\varepsilon_{f}$. In this context, the engineering stress is defined as $\sigma_{eng}=P/A_{0}$, where *P* is the instantaneous load during the test and $A_{0}$ is the initial transversal area computed as $w_{0}t_{0}$, where $w_{0}$ is the initial width (at the center of the sample) and $t_{0}$ is the initial thickness. Moreover, the axial engineering strain is defined as $\varepsilon_{eng}=\left(L-L_{0}\right)/L_{0}$, where *L* and $L_{0}$ are the current and initial extensometer lengths, respectively.

Assuming volume invariance and homogeneous stress and strain patterns before the necking formation, the true stress (which is equal to the equivalent stress $\bar{\sigma }$ for this specific condition), true (logarithmic) strain, and equivalent plastic strain are, respectively, defined as $\sigma =\sigma_{eng}\left(1+\varepsilon_{eng}\right)$, $\varepsilon =ln(1+\varepsilon_{eng})$, and ${\bar{\varepsilon }}^{p}=\varepsilon -\frac{\sigma }{E}$. According to the works in [37,38], the equivalent stress in terms of the equivalent plastic strain curve should be used to characterize the material hardening response.

To characterize the plastic anisotropy, use is made of the Lankford coefficients which are defined as $R_{\theta }=\frac{{\dot{\varepsilon }}_{w}}{{\dot{\varepsilon }}_{t}}$, where θ is the angle with respect to the rolling direction and the true strain rates ${\dot{\varepsilon }}_{w}$ and ${\dot{\varepsilon }}_{t}$ are, respectively, related to the width and thickness evolutions of the sample. Once again, assuming an isochoric and homogeneous plastic flow, ${\dot{\varepsilon }}_{t}=-\left({\dot{\varepsilon }}_{l}+{\dot{\varepsilon }}_{w}\right)$, where ${\dot{\varepsilon }}_{l}$ is the true strain rate along the axial direction of the sample. The true strain rates ${\dot{\varepsilon }}_{w}$ and ${\dot{\varepsilon }}_{l}$ are calculated on the basis of the displacement field obtained by DIC at some specific points located within the extensometer length.

#### 2.1.3. Bulge Test

The hydraulic bulge test consists in applying a pressure to a circular section of the sample whose perimeter circumference is clamped. The sample used was a 300 × 300 mm square with the dimensions shown in Figure 3a. The height of the dome and the applied pressure evolutions are measured during the test by means of a comparing dial gauge (resolution of 0.01 mm with a range of 40 mm) and an analog manometer (resolution of 5 bar with a range of 0 to 100 bar), respectively. The test setup involved an image recorder to register the motion of the dial gauges. A manual hydraulic cylinder was used to generate the pressure in the bottom chamber. The set-up is shown in Figure 3b.

The hydraulic bulge test is performed to assess the biaxial response of the sheet. As the material is anisotropic, the classical analytical equations of this test, in which the assumed isotropy leads to an equibiaxial state, are not adequate in this case. Therefore, as shown in Section 3.2, numerical simulations have been carried out in order to more accurately predict the stress and strain patterns that develop during this test.

### 2.2. Constitutive Modelling

To describe the mechanical response of the copper sheet, a large strain plasticity model is used that respectively assumes isotropy and orthotropy for the elastic and plastic regimes. The stress–strain law is given by [37]
(1)$$\sigma =C:(e-e^{p})$$
where σ is the Cauchy stress tensor, C is the elastic constitutive tensor, and e is the Almansi strain tensor with its plastic component $e^{p}$.

The adopted yield criterion is the Hill-48 function defined as
(2)$$f=F{\left(\sigma_{yy}-\sigma_{zz}\right)}^{2}+G{\left(\sigma_{xx}-\sigma_{zz}\right)}^{2}+H{\left(\sigma_{yy}-\sigma_{xx}\right)}^{2}+2L\sigma_{yz}^{2}+2M\sigma_{xz}^{2}+2N\sigma_{xy}^{2}-{C^{p}}^{2}={\bar{\sigma }}^{2}-{C^{p}}^{2}=0$$
where the subscripts of σ denoted by *x*, *y*, and *z* are, respectively, related to RD, TD, and ND shown in Figure 1. Moreover, *F*, *G*, *H*, *L*, *M*, and *N* are the Hill-48 parameters; $C^{p}$ is the hardening function; and $\bar{\sigma }$ is the equivalent (i.e., Hill-48) stress. Because of the difficulty to estimate the out-of-plane shear parameters *L* and *M* with the available instruments, it is assumed that they are equal to the in-plane shear parameter *N*. The expressions for each parameter as a function of the yield limits associated to the tensile (with samples cut along RD, DD and TD) and the biaxial (denoted by subscript B) tests are [39]:(3)$$F=\frac{1}{2}\left(\frac{\sigma_{RD}^{2}}{\sigma_{TD}^{2}}+\frac{\sigma_{RD}^{2}}{\sigma_{B}^{2}}-1\right)$$
(4)$$G=\frac{1}{2}\left(1+\frac{\sigma_{RD}^{2}}{\sigma_{B}^{2}}-\frac{\sigma_{RD}^{2}}{\sigma_{TD}^{2}}\right)$$
(5)$$H=\frac{1}{2}\left(1+\frac{\sigma_{RD}^{2}}{\sigma_{TD}^{2}}-\frac{\sigma_{RD}^{2}}{\sigma_{B}^{2}}\right)$$
(6)$$N=\frac{1}{2}\left(4\frac{\sigma_{RD}^{2}}{\sigma_{DD}^{2}}-\frac{\sigma_{RD}^{2}}{\sigma_{B}^{2}}\right)=L=M.$$

The hardening function used in this study is a modified Voce law [40] written for RD as
(7)$$C^{p}=\sigma_{RD}+K\varepsilon^{p}+\left(\sigma_{sat}-\sigma_{RD}\right)\left[1-exp\left(-n{\bar{\varepsilon }}^{p}\right)\right]$$
where *K*, $\sigma_{sat}$, and *n* are hardening parameters, and ${\bar{\varepsilon }}^{p}$ is the effective plastic strain whose rate is given below. Moreover, in order to adequately represent the strain corresponding to the onset of necking formation $\varepsilon_{neck}$, Equation (Equation 7) is complemented, as proposed in [41], with the following definition of the parameter *K*,
(8)$$K=\frac{\sigma_{sat}-\left(\sigma_{sat}-\sigma_{RD}\right)exp\left(-n\varepsilon_{neck}\right)}{1-\varepsilon_{neck}}.$$

According to the results to be presented in Section 3.1, the following non-associated flow rule is chosen to properly capture the plastic response of the material,
(9)$$L_{\nu }\left(e^{p}\right)=\dot{\lambda }\frac{\partial g}{\partial \sigma }$$
where $L_{\nu }$ is the well-known Lie (frame-indifferent) derivative, $\dot{\lambda }$ is the plastic consistency parameter (computed according to classical concepts of the plasticity theory), and *g* is the flow potential chosen in this work as the Hill-48 function,
(10)$$g=F^{\prime }{\left(\sigma_{yy}-\sigma_{zz}\right)}^{2}+G^{\prime }{\left(\sigma_{xx}-\sigma_{zz}\right)}^{2}+H^{\prime }{\left(\sigma_{yy}-\sigma_{xx}\right)}^{2}+2L^{\prime }\sigma_{yz}^{2}+2M^{\prime }\sigma_{xz}^{2}+2N^{\prime }\sigma_{xy}^{2}$$
but with parameters defined in terms of the Lankford coefficients (obtained from samples tested at RD, DD, and TD) as [42]:(11)$$F^{\prime }=\frac{R_{RD}}{R_{TD}\left(1+R_{RD}\right)}$$
(12)$$G^{\prime }=\frac{1}{1+R_{RD}}$$
(13)$$H^{\prime }=\frac{R_{RD}}{1+R_{RD}}$$
(14)$$N^{\prime }=\frac{1}{1+R_{RD}}\left(\frac{R_{RD}}{R_{TD}}+1\right)\left(R_{DD}+\frac{1}{2}\right)=L^{\prime }=M^{\prime }$$

Finally, the rate of ${\bar{\varepsilon }}^{p}$ is given in this context by ${\dot{\bar{\varepsilon }}}^{p}=\frac{\sigma :L_{\nu }\left(e^{p}\right)}{\bar{\sigma }}$ [18,19].

### 2.3. Numerical Simulations

Numerical simulations of the tensile and bulge tests are, respectively, used to fit and validate the obtained results with the corresponding experimental measurements. All the computations are performed with the model described in Section 2.2 via an in-house finite element code extensively validated in many forming problems [37]. The finite element meshes are composed of linear hexahedral elements with a B-bar strain–displacement matrix to prevent the volumetric locking effect on the numerical solution when incompressible plastic flows are considered [37]. Moreover, a mesh refinement sensitive analysis was previously carried out in order to ensure mesh-independent results. For both problems, 4 elements along the thickness of the sheet were used.

The boundary conditions and symmetry planes considered in the simulation of the tensile test for samples cut along RD and TD are shown in Figure 4a while those of sample DD are depicted in Figure 4b. The same mesh is used for the RD and TD samples (obviously exchanging both directions for each case) where only one eighth of the geometry is discretized due to the existence of three symmetry planes in these cases. This last condition is not fulfilled in the DD sample for which only one half of the domain is discretized owing to the symmetry plane whose normal is ND. In the three samples, the axial displacement at the boundary is imposed up to the corresponding value of the experimental displacement at the fracture stage for each case.

The boundary conditions applied in the simulation of the bulge test are shown in Figure 5. In this case, only one fourth of the domain is discretized as two symmetry planes are considered. While the outer edges of the sample are constrained, an increasing pressure is applied on the bottom surface of the sheet until a dome vertical displacement value similar to that achieved experimentally is reached.

### 2.4. Fitting Procedure

A three-step procedure based on the minimization, via the Levenberg–Marquardt algorithm [43], of the error between the experimental tensile test measurements and the corresponding analytical–numerical predictions is proposed to derive the constitutive material parameters of the constitutive model presented in Section 2.2. The error for each step is quantified by means of the normalized root mean square deviation (NRMSD), which is given by
(15)$$\mathrm{NRMSD}=\frac{1}{\Delta }\sqrt{\frac{1}{m}\sum_{i=1}^{m}{\left(\hat{y_{i}}-y_{i}\right)}^{2}}$$
where *m* is the number of experimental data, $y_{i}$ is the experimental measurement, $\hat{y_{i}}$ is the analytical–numerical fitted value and $\Delta =|y_{max}-y_{min}|$ is the interval width bounded by the maximum and minimum values of $y_{i}$.

Step 1.

Hardening parameters derived from the RD tensile curve.

The fitting is carried out using the experimentally measured equivalent stress-equivalent plastic strain curve for the RD tensile sample and the analytical expressions given by Equation (Equation 7) together with the constraint imposed by Equation (Equation 8). Considering that $\sigma_{RD}$ and $\varepsilon_{neck}$ are obtained from the experimental RD tensile curve, the resulting hardening parameters to be calibrated are $\sigma_{sat}$ and *n*.

Step 2.

First prediction of the Hill-48 parameters based on the Lankford coefficients.

Since the biaxial yield limit $\sigma_{B}$ is not available, the yield function parameters given by Equations (Equation 3)–(Equation 6) cannot be directly computed. Therefore, an associated plastic model is preliminary adopted as a first approximation to describe the material response (it should be mentioned that non-associated model will be used in Step 3 for the final parameter identification). In this context, the sheet orthotropy is evaluated by means of the Lankford coefficients along RD, DD, and TD assuming an isochoric strain field whose in-plane components were obtained from the DIC measurements within the interval of homogeneous strain, i.e., before the onset of necking formation. Then, Equations (Equation 11)–(Equation 14) are used to compute the initial values of the Hill-48 parameters. The model response using these parameters is assessed in the $\varepsilon_{w}$–$\varepsilon_{l}$ (true strains along the axial and width directions of the sample, respectively) and $\sigma_{eng}$–$\varepsilon_{eng}$ curves for the whole deformation range up to the rupture stage. In this work, the $\varepsilon_{w}$–$\varepsilon_{l}$ curve is plotted at the central point of the neck in order to achieve a larger strain interval and, thus, improve the prediction capability of the calibration methodology.

Step 3.

Correction of the parameters of the flow and yield potential functions.

In general, the model response considering the Hill-48 parameters analytically obtained in Step 2 does not accurately adjust the corresponding experimental measurements along the whole test in which a triaxial and non-homogeneous stress and strain patterns develop after the early necking formation occurring in the tensile samples. Therefore, these parameters need to be amended. To this end, the proposed correction phase is performed via numerical simulations of the tensile test for the different samples in two successive sub-steps detailed below. This correction process is repeated until the error NRMSD for each curve is minimized.

3.1. Correction of the flow potential function parameters $F^{\prime }$, $G^{\prime }$, $H^{\prime }$, and $N^{\prime }$ using the experimental and computed $\varepsilon_{w}$–$\varepsilon_{l}$ curves. Each curve allows the determination of specific parameters keeping the rest constant, i.e., the correction sequence is (a) parameters $G^{\prime }$ and $H^{\prime }$ with the RD curve subjected to the condition $G^{\prime }+H^{\prime }=1$ (which is also fulfilled by Equations (Equation 12) and (Equation 13)), (b) parameter $F^{\prime }$ with the TD curve, and (c) parameter $N^{\prime }$ with the DD curve.

3.2. Correction of the yield function parameters *F*, *G*, *H*, and *N* using the experimental and computed $\sigma_{eng}$–$\varepsilon_{eng}$ curves. Once again, each curve allows the determination of specific parameters keeping the rest constant, i.e., the correction sequence is (a) parameters *G* and *H* assuming $G=G^{\prime }$ and $H=H^{\prime }$ (it should be noted that these two parameters are not corrected with the RD curve since it has been considered as a reference data to calibrate the hardening response and, therefore, *G* and *F* do not play any role provided G+H=1 is preserved as also stated by Equations (Equation 4) and (Equation 5), (b) parameter *F* with the TD curve, and (c) parameter *N* with the DD curve.

## 3. Results

### 3.1. Tensile Test

The average experimental engineering stress–strain curves for the RD, DD, and TD tensile samples are shown in Figure 6, where marked differences in the material response can be seen for the different in-plane directions of the sheet. These data allow obtaining the elastic Young modulus *E*, the yield strength $\sigma_{y}$, the ultimate tensile strength $\sigma_{uts}$, the strain at the onset of necking formation $\varepsilon_{neck}$, the strain at the fracture stage $\varepsilon_{f}$, and the Lankford coefficients as shown in Table 1.

According to Step 1 of Section 2.4, Figure 7 shows the fitting of the hardening function with the tensile experimental data for the RD sample. In order to guarantee a homogeneous strain pattern in the sample, this procedure is applied within a strain range up to the value of necking formation ${\bar{\varepsilon }}^{p}=6.0\%$, which is equivalent to the average engineering strain of $\varepsilon_{neck}=6.5\%$ (see Table 1). Taking the value of $\sigma_{RD}$ as the average yield strength of Table 1, the rest of the derived hardening parameters are summarized in Table 2.

The average values of the Lankford coefficients included in Table 1 were computed at the corresponding strain level $\varepsilon_{neck}$ assuming, as already mentioned in Step 2 of Section 2.4, an isochoric strain field whose in-plane components were measured by DIC. These Lankford coefficients are used to obtain a first approximation of the Hill-48 parameters of the preliminary adopted associated plastic model. These parameters, summarized in Table 3, are considered in turn to simulate the tensile test from which Figure 8 and Figure 9 show for the three sample directions the experimental and computed $\varepsilon_{w}$–$\varepsilon_{l}$ (at the central point of the neck) and $\sigma_{eng}$-$\varepsilon_{eng}$ curves, respectively. Although Figure 8 exhibits a reasonable agreement between the experimental and numerical results, this is not the case for the DD and TD results plotted in Figure 9. This fact leads, as stated in Section 2.4, to the need to recalculate the Hill-48 parameters.

Then, Step 3 of the calibration procedure described in Section 2.4 is applied through numerical simulations of the tensile tests to obtain a more representative description of the material behavior in the plastic regime. The resulting parameters of the yield and flow potential functions are included in Table 3. As shown in Table 4, the numerical predictions obtained with these parameters minimize the experimental–numerical error for the results plotted in Figure 8 and Figure 9. This is, in particular, more apparent for the DD and TD stress–strain curves, respectively, plotted in Figure 9b,c which, as already mentioned, exhibit the largest initial NRMSD values.

Finally, Figure 10 shows the experimentally measured via DIC and computed axial strain ($\varepsilon_{l}$) at two levels of axial engineering strain ($\varepsilon_{eng}$) for the three tensile sample directions (RD, DD, and TD). For these orientations, the neck occurs at very different levels of strain. This is the reason why different scales are defined for each orientation of the specimen. The level 4% is used to visualize strain gradients that appear at low strain levels, such as those exhibited by the DD and TD specimens, where the level 8% is chosen to visualize the same effect in the RD orientation.

### 3.2. Bulge Test

Figure 11 shows the average experimental and computed pressure-dome height curve in the bulge test. The two plotted numerical predictions have been respectively obtained at the end of Steps 2 and 3 of the fitting procedure described in Section 2.4, i.e., using the Hill-48 parameters presented in Table 3.

For the maximum dome height of the sample, Figure 12 depicts an experimental–numerical comparison of the thickness profiles along the RD and TD radial lines while Figure 13 shows the computed in-plane maximum and minimum principal stresses at the outer surface. These results clearly illustrate the noticeable orthotropic response exhibited by the material at the end of the test.

## 4. Discussion

An important aspect that has been taken into account in the proposed mechanical characterization methodology applied to the studied material is, as shown in Figure 6 and Table 1, the early necking formation that causes, more markedly in the DD and TD samples, non-homogeneous stress and strain patterns. Due to this, the hardening and orthotropic parameters cannot be exclusively derived by means of analytical expressions but through numerical simulations able to describe the complex triaxial state that is generated in the post-necking regime.

Figure 7 shows an excellent fitting (with NRMSD <0.01) obtained with the modified Voce law of the material response along the whole deformation range exhibited by the RD sample. This would not be the case if the Voce or Swift laws [10] had been used, for which the hardening would be under or overestimated, respectively. Moreover, it should be noted that this good agreement with the experimental data was achieved thanks to the restriction imposed on the hardening parameter *K* (see Equation (Equation 8)) as, if this condition had not been taken into account, the strain corresponding to the onset of necking formation would have been unrealistically predicted as 30% instead of the experimentally measured value of 6.5%.

The results plotted in Figure 8 and Figure 9, and to a lesser extent the ultimate tensile strength ($\sigma_{uts}$) shown in Table 1, clearly show the orthotropic and non-associated nature of the material plastic response (this is consequently reflected in the final Hill-48 parameters shown in Table 3 that differ from those related to an isotropic and associated response, i.e., $F=F^{\prime }=G=G^{\prime }=H=H^{\prime }=0.5$ and $N=N^{\prime }=1.5$). They also reveal the soundness of the proposed fitting procedure in properly describing the stress and strain patterns that develop during practically the entire deformation range of the samples along the different studied directions. As the hardening model is adjusted in the RD direction, the numerical results agree well with the experimental measurements in the whole range of strain for this direction. However, some differences between the experimental and numerical results are observed for DD and TD, which are mainly manifested at the higher strain levels. Therefore, it may be possibly necessary to include more deformation paths in the characterization to assess the choice of the yield function to improve the model predictions. Moreover, it should be also noted that the different strain levels at which the neck is formed for the RD, DD, and TD samples (Table 1) is reasonably reproduced by the model.

The strain contours plotted in Figure 10 exhibit an overall good agreement between the experimental and computed results. Although the major discrepancies are related to the maximum values of the highly localized strain field, the extension of the necking zone is, however, notably well captured by the numerical predictions. This confirms the reasonable predictive capabilities achieved by the constitutive model used in this work when its parameters are properly identified.

The performance of the previous mechanical characterization is evaluated in the analysis of the bulge test in which the material is mainly subjected to biaxial loading conditions [7,44]. It is seen that the computed pressure–dome height curve and thickness profiles, respectively, shown in Figure 11 and Figure 12 are in good agreement with the corresponding experimental data. Although no significant differences are observed for the numerical predictions computed with the material parameters derived in Steps 2 and 3 of the fitting procedure described in Section 2.4, a better experimental–numerical agreement is obtained by the results from the non-associate model. Finally, Figure 13 clearly shows the non–equibiaxial stress pattern that develops during the test due to the already mentioned noticeable orthotropy of the material, where, according to the hardening responses shown in Figure 9, the peak values of the maximum and minimum principal stress contours are, respectively, aligned with RD and TD.

## 5. Conclusions

An elastoplastic characterization of a rolled C11000-H2 99.90% pure copper sheet considering the orthotropic non-associated Hill-48 criterion together with a modified Voce hardening law has been presented. To this end, a robust inverse calibration approach, based on an experimental–analytical–numerical iterative predictor–corrector methodology, has been proposed and used with tensile test measurements of samples stretched along three in-plane directions. The main difficulty related to the early development of non-homogeneous stress and strain patterns has been successfully tackled with this approach as the calibration procedure carried out in this work was found to adequately describe the orthotropic behavior of the material during the whole deformation range up to the rupture stage. Besides, such mechanical characterization was also experimentally validated in the simulation of the bulge test where sound predictions of the pressure–dome height evolution and final thickness profiles have been achieved.

Finally, future research including more strain paths will be devoted to improve the current material characterization by assessing the effects of out-plane shear response, kinematic hardening and mechanical damage and, in addition, to explore the potential benefits of using more advanced yield criteria.

## Figures and Tables

**Figure 1 materials-13-05193-f001:**
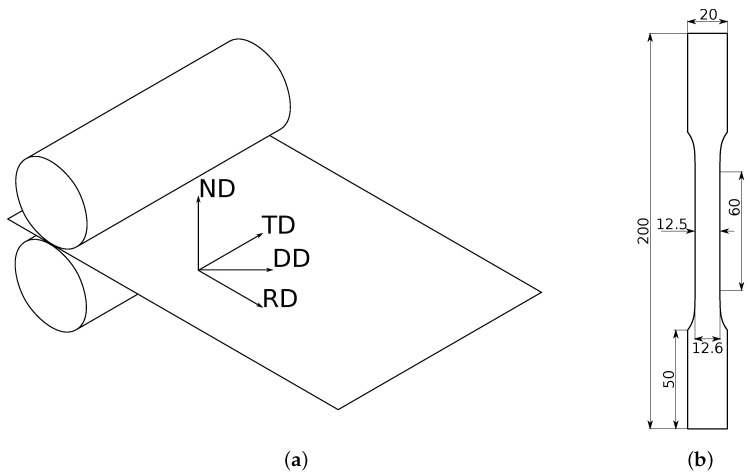
(**a**) Common directions of laminated sheets and (**b**) dimensions of the tensile specimen (in mm).

**Figure 2 materials-13-05193-f002:**
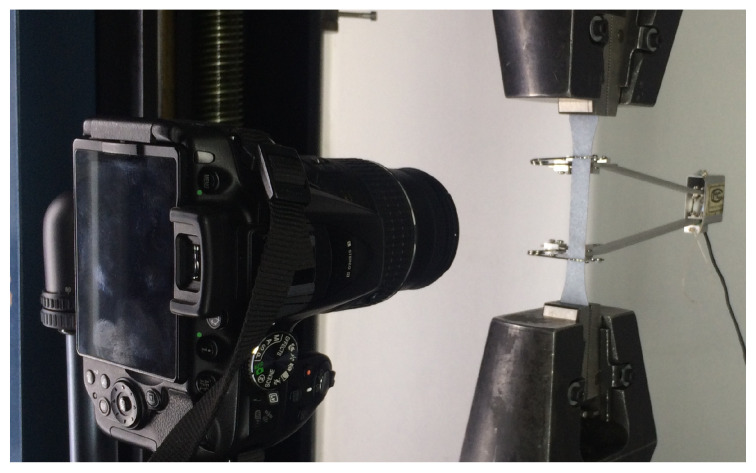
Tensile test set-up.

**Figure 3 materials-13-05193-f003:**
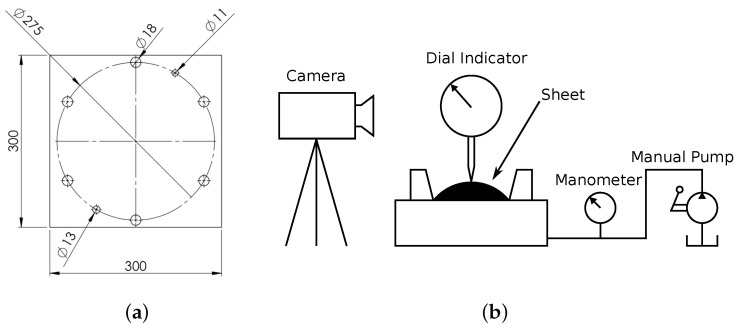
Bulge test (**a**) specimen (dimensions in mm) and (**b**) set-up.

**Figure 4 materials-13-05193-f004:**
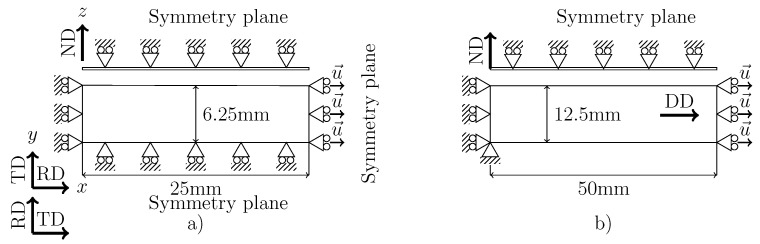
Tensile test boundary conditions for the (**a**) RD and TD and (**b**) DD specimens (side and top views).

**Figure 5 materials-13-05193-f005:**
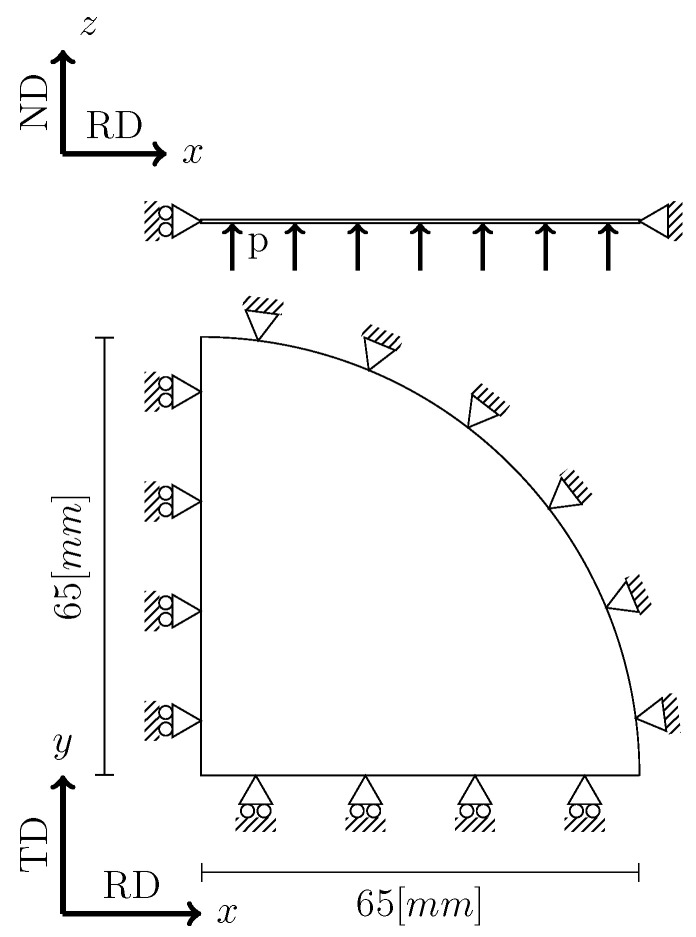
Bulge test boundary conditions (side and top views).

**Figure 6 materials-13-05193-f006:**
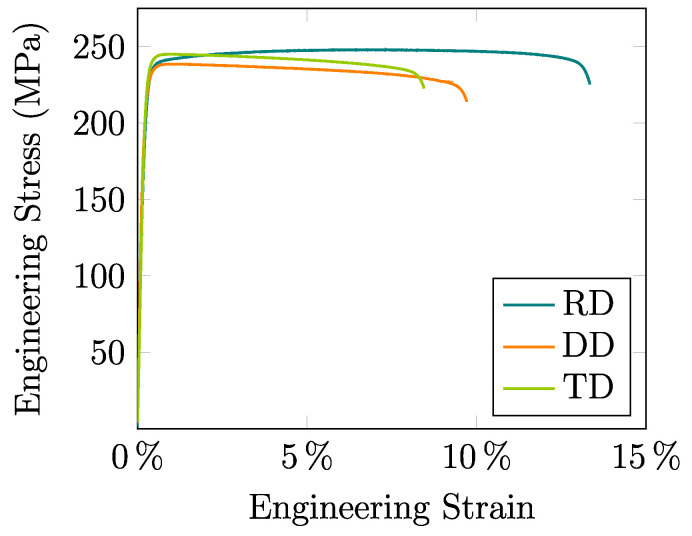
Average experimental engineering stress–strain curves for the RD, DD, and TD tensile samples obtained after three tests per direction.

**Figure 7 materials-13-05193-f007:**
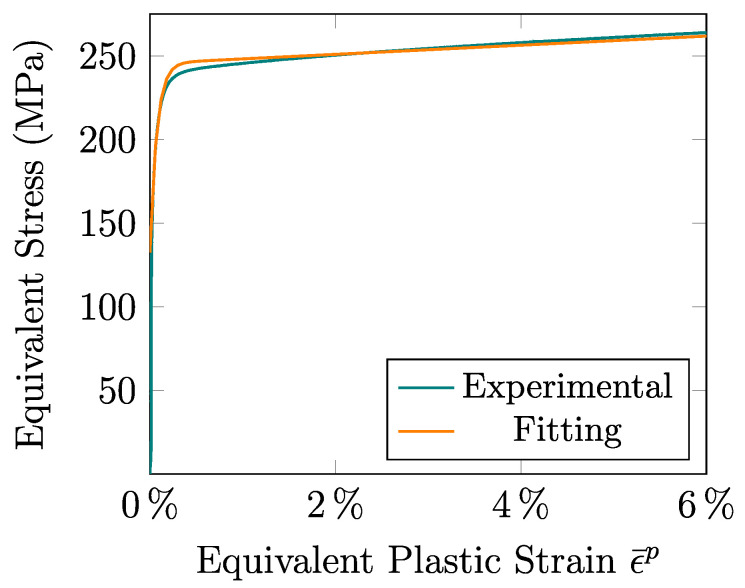
Fitting of the hardening parameters from the average experimental equivalent stress–equivalent plastic strain curve for the RD tensile sample.

**Figure 8 materials-13-05193-f008:**
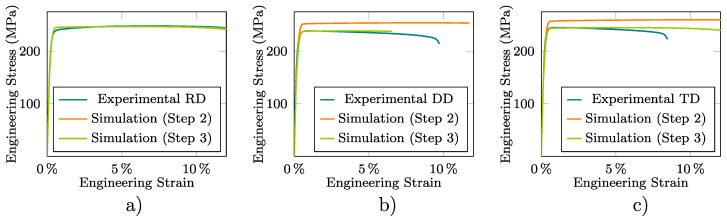
Fitting of the Hill-48 yield and flow potential functions from the average experimental $\varepsilon_{w}$–$\varepsilon_{l}$ curve for the (**a**) RD, (**b**) DD, and (**c**) TD tensile samples at the central point of the neck.

**Figure 9 materials-13-05193-f009:**
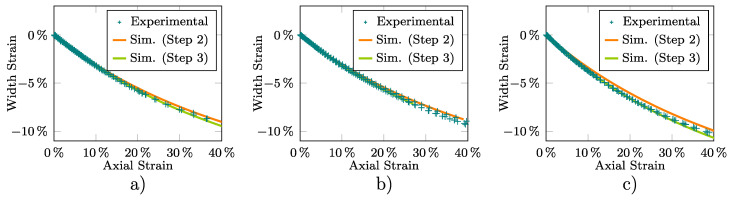
Fitting of the Hill-48 yield and flow potential functions from the average experimental $\sigma_{eng}$-$\varepsilon_{eng}$ curve for the (**a**) RD, (**b**) DD, and (**c**) TD tensile samples.

**Figure 10 materials-13-05193-f010:**
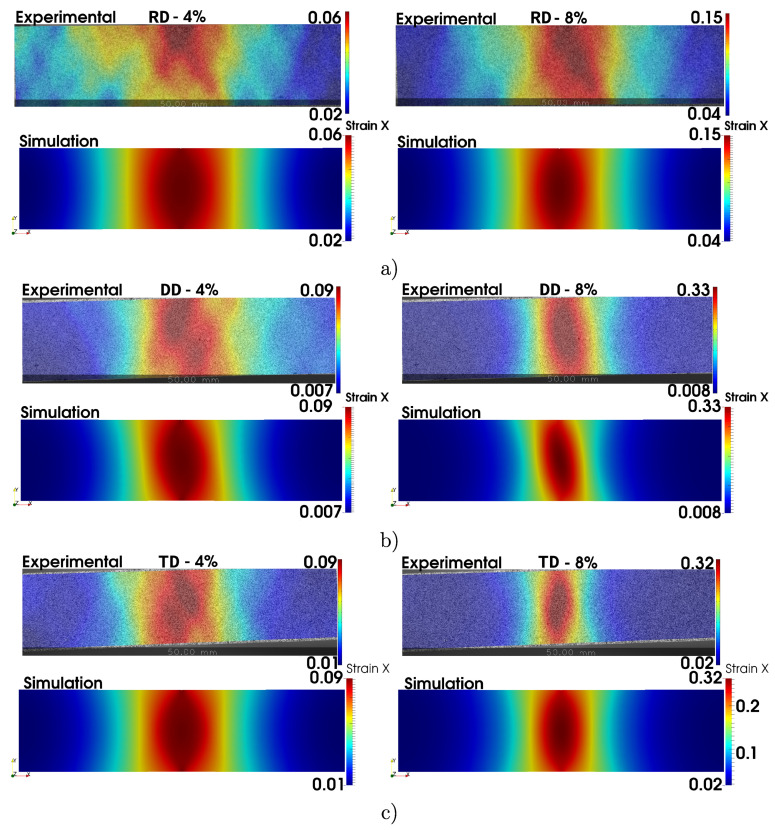
Comparison of DIC measurements and computed $\varepsilon_{l}$ contours for 4% and 8% of $\varepsilon_{eng}$ for the (**a**) RD, (**b**) DD, and (**c**) TD tensile samples.

**Figure 11 materials-13-05193-f011:**
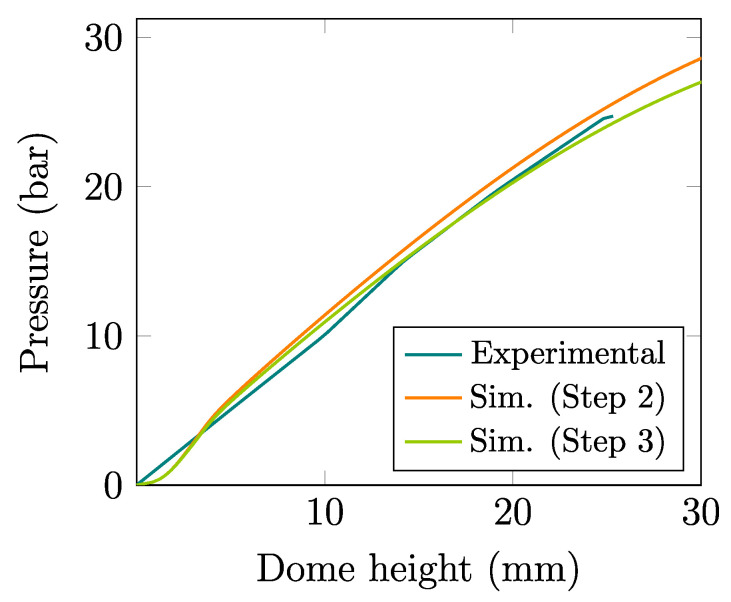
Average experimental, obtained after three tests, and computed pressure–dome height curve in the bulge test.

**Figure 12 materials-13-05193-f012:**
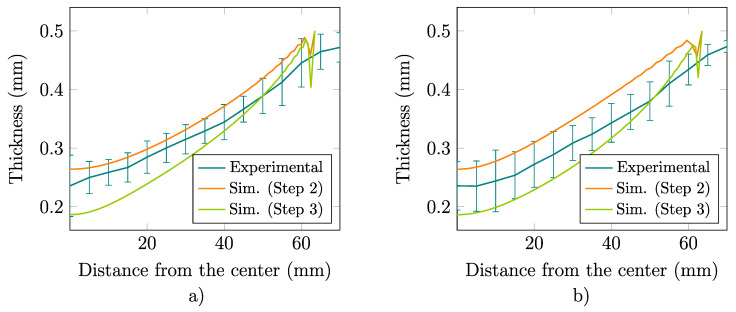
Experimental and computed thickness profiles for the maximum dome height along the radial (**a**) RD and (**b**) TD lines of the bulge sample.

**Figure 13 materials-13-05193-f013:**
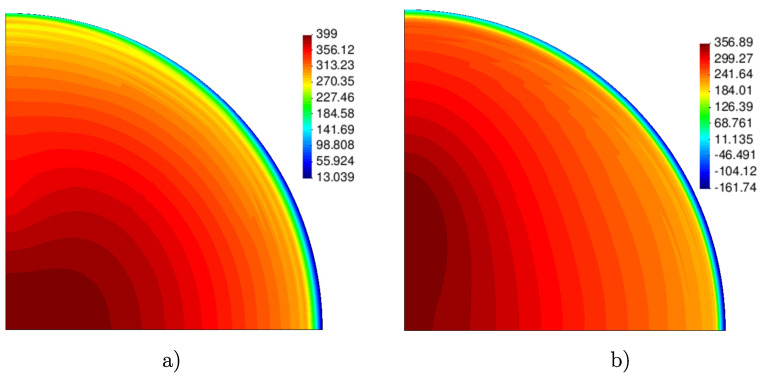
Computed in-plane (**a**) maximum and (**b**) minimum principal stress contours (in MPa) at the outer surface of the bulge sample for the maximum dome height (RD is the horizontal direction).

**Table 1 materials-13-05193-t001:** Experimentally measured values for the RD, DD, and TD tensile samples.

Direction	E (GPa)	$\sigma_{y}$ (MPa)	$\sigma_{uts}$ (MPa)	$\varepsilon_{neck}$ [%]	$\varepsilon_{f}$ [%]	*R*
RD	108±8.8	132.0±2.3	248.1±0.1	6.50±0.50	13.4±0.10	0.56±0.03
DD	110±13.3	128.1±13.8	238.7±1.0	0.95±0.08	9.7±0.27	0.59±0.06
TD	108±11.3	135.0±2.0	245.1±0.5	0.90±0.03	8.8±0.29	0.60±0.04

± Standard deviation.

**Table 2 materials-13-05193-t002:** Parameters of the hardening function.

$\sigma_{RD}$ (MPa)	*K* (MPa)	$\sigma_{sat}$ (MPa)	*n*
132.0	272.7	245.4	1353

**Table 3 materials-13-05193-t003:** Parameters of the Hill-48 yield and flow potential functions.

Parameter	*F/F’*	*G/G’*	*H/H’*	*N/N’*
Initial values for *f* and *g* (Step 2)	0.550	0.640	0.360	1.300
Final values for *f* (Step 3)	0.650	0.625	0.375	1.500
Final values for *g* (Step 3)	0.480	0.625	0.375	1.230

**Table 4 materials-13-05193-t004:** Normalized root mean square deviation (NRMSD) in the $\varepsilon_{w}$–$\varepsilon_{l}$ and $\sigma_{eng}$-$\varepsilon_{eng}$ curves.

Curve	Initial NRMSD (Step 2)	Final NRMSD (Step 3)
$\varepsilon_{w}$–$\varepsilon_{l}$ for RD	0.021	0.020
$\varepsilon_{w}$–$\varepsilon_{l}$ for DD	0.021	0.016
$\varepsilon_{w}$–$\varepsilon_{l}$ for TD	0.050	0.007
$\sigma_{eng}$-$\varepsilon_{eng}$ for RD	0.020	0.019
$\sigma_{eng}$-$\varepsilon_{eng}$ for DD	0.160	0.051
$\sigma_{eng}$-$\varepsilon_{eng}$ for TD	0.130	0.049

## Data Availability

The raw/processed data required to reproduce these findings cannot be shared at this time as the data also forms part of an ongoing study.

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
