# Peer review of "Mechanical Characterization of the Elastoplastic Response of a C11000-H2 Copper Sheet"

_materials, 2020, doi:10.3390/ma13225193_

Round 1

Reviewer 1 Report

Figures and tables are not positioned close to where they are mentioned in the text. Some figures are positioned even in different chapters from those in which they are referred.  It is very difficult to follow the reasoning of the paper because of this.

The presentation of the results is not well argued by the explanations of the figures invoked in this chapter.

Figure 10 require additional explanations for understanding the different values indicated (0.06 and 0.15 by example). It would have been more useful to see completely the legend of the axial strain values in order to understand (by analyzing the color areas) the axial strain distribution in the observed area.

It would be useful to specify the value of the distance between the experimental curve and that obtained by numerical simulation at different step.

Due to the incorrect positioning of the figures, a sentence starts on page 14 and ends on page 18 (situation marked by me in the text of the paper).

Figure 9 shows that Step 3 of the methodology generates a larger difference between the experimental curve and the one obtained by numerical simulation, for DD and TD. How is this explained? The Discussion and Conclusions chapters do not seem to take into account these differences.  

Author Response

Revised manuscript “Mechanical characterization of the elastoplastic response of a C11000-H2 copper sheet”, submitted to Materials (manuscript materials-979036).

The authors thank the valuable comments of the reviewer. The changes in the revised version of this article, which were made according to the reviewer’s comments, are highlighted in red in the manuscript.

Reviewer 1

Comments and Suggestions for Authors

Figures and tables are not positioned close to where they are mentioned in the text. Some figures are positioned even in different chapters from those in which they are referred.  It is very difficult to follow the reasoning of the paper because of this.

Authors’ response: Figures and Tables of the different Sections were correctly positioned.

The presentation of the results is not well argued by the explanations of the figures invoked in this chapter.

Authors’ response: Improvements in both the presentation and discussion of results are included in Sections 3 and 4.

Figure 10 require additional explanations for understanding the different values indicated (0.06 and 0.15 by example). It would have been more useful to see completely the legend of the axial strain values in order to understand (by analyzing the color areas) the axial strain distribution in the observed area.

Authors’ response: An explanation related to the values chosen in Figure 10 is included in Section 3.

“Finally, Figure 10 shows the experimentally measured via DIC and computed axial strain ( ) at two levels of axial engineering strain ( ) for the three tensile sample directions (RD, DD and TD). For these orientations, the neck occurs at very different levels of strain. This is the reason why different scales are defined for each orientation of the specimen. The level 4% is used to visualize strain gradients that appear at low strain levels, such as those exhibited by the DD and TD specimens, where the level 8% is chosen to visualize the same effect in the RD orientation.”

It would be useful to specify the value of the distance between the experimental curve and that obtained by numerical simulation at different step.

Authors’ response: The distance between the experimental and numerical curves was assessed through the calculation of the error defined in equation (9). The values associated to different results (i.e., width strain vs axial strain and stress vs strain for the three sample directions) are summarized in Table 4.

Due to the incorrect positioning of the figures, a sentence starts on page 14 and ends on page 18 (situation marked by me in the text of the paper).

Authors’ response: Amended.

Figure 9 shows that Step 3 of the methodology generates a larger difference between the experimental curve and the one obtained by numerical simulation, for DD and TD. How is this explained? The Discussion and Conclusions chapters do not seem to take into account these differences.  

Authors’ response: An explanation related to this aspect is included in Section 4.

“Since the hardening model is adjusted in the RD direction, the numerical results agree well with the experimental measurements in the whole range of strain for this direction. However, some differences between the experimental and numerical results are observed for DD and TD, which are mainly manifested at the higher strain levels. Therefore, it may be possibly necessary to include more deformation paths in the characterization to assess the choice of the yield function to improve the model predictions.”

Additional comments: The methods, presentation of results and conclusions were revised and improved.

Reviewer 2 Report

The paper presents a fully identification procedure for the elasto-plastic characterization of a rolled C11000-H2 pure copper sheet. The elasto-plastic parameters of this material are difficult to identify, due to the early development of plastic strains (< 1%). Despite this difficulty, the authors have successfully identified the isotropic hardening parameters as well as the Hill’48 anisotropic coefficients.

Despite that many works exist in the literature on the same topic, this study deserves to be published because of the difficulty of the material parameters within such low strain level. However, the following minor revision has to be addressed before the final publication:

  • The scientific English of the paper has to be greatly improved. There are several grammatical errors and spelling mistakes within the paper. The content of the paper could be significantly improved if it is revised by a proofreader with good written English skills.
  • The references are not cited in the order of their appearance within the text.
  • Page 8: Hexahedral elements are used in the simulations. What is the order (linear/quadratic).
  • Page 14: paragraph “Finally, …”, please correct the sentence including “axial axial”.
  • 12: please use the same Y-axis limits between both figures.
  • Section Discussion: the first sentence is too long. Please revise it and make it clearer.

Author Response

Revised manuscript “Mechanical characterization of the elastoplastic response of a C11000-H2 copper sheet”, submitted to Materials (manuscript materials-979036).

The authors thank the valuable comments of the reviewer. The changes in the revised version of this article, which were made according to the reviewer’s comments, are highlighted in red in the manuscript.

Reviewer 2

Comments and Suggestions for Authors

The paper presents a fully identification procedure for the elasto-plastic characterization of a rolled C11000-H2 pure copper sheet. The elasto-plastic parameters of this material are difficult to identify, due to the early development of plastic strains (< 1%). Despite this difficulty, the authors have successfully identified the isotropic hardening parameters as well as the Hill’48 anisotropic coefficients.

Despite that many works exist in the literature on the same topic, this study deserves to be published because of the difficulty of the material parameters within such low strain level. However, the following minor revision has to be addressed before the final publication:

  • The scientific English of the paper has to be greatly improved. There are several grammatical errors and spelling mistakes within the paper. The content of the paper could be significantly improved if it is revised by a proofreader with good written English skills.

Authors’ response: The use of the English language was revised throughout the manuscript.

  • The references are not cited in the order of their appearance within the text.

Authors’ response: Amended.

  • Page 8: Hexahedral elements are used in the simulations. What is the order (linear/quadratic).

Authors’ response: Linear hexahedral elements with the B-bar technique are used. This is included in Section 2.3.

  • Page 14: paragraph “Finally, …”, please correct the sentence including “axial axial”.

Authors’ response: Amended.

  • 12: please use the same Y-axis limits between both figures.

Authors’ response: The same limits are now considered in Figure 12.

  • Section Discussion: the first sentence is too long. Please revise it and make it clearer.

Authors’ response: Amended.

Additional comments: The methods, presentation of results and conclusions were revised and improved.